# Antituberculosis Drug Interactions with Membranes: A Biophysical Approach Applied to Bedaquiline

**DOI:** 10.3390/membranes9110141

**Published:** 2019-10-30

**Authors:** Marina Pinheiro, Heinz Amenitsch, Salette Reis

**Affiliations:** 1LAQV, REQUIMTE, Departamento de Ciências Químicas, Faculdade de Farmácia, Universidade do Porto, 4050-313 Porto, Portugal; shreis@ff.up.pt; 2Institute of Inorganic Chemistry, Graz University of Technology, Stremayergasse 6/V, 8010 Graz, Austria; heinz.amenitsch@elettra.eu

**Keywords:** antibiotic, liposomes, membrane models, WAXS, SAXS, tuberculosis

## Abstract

This work focuses on the interaction of the novel and representative antituberculosis (anti-TB) drug bedaquiline (BDQ) with different membrane models of eukaryotic and prokaryotic cells. The effect of BDQ on eukaryotic cell membrane models was assessed using liposomes, namely, multilamellar vesicles (MLVs) made of 1,2-dimyristoyl-*rac*-glycero-3-phosphocholine (DMPC) and also a mixture of DMPC and cholesterol (CHOL) (8:2 molar ratio). To mimic the prokaryotic cell membrane, 1,2-dimyristoyl-*sn*-glycero-3-phospho-*rac*-(1-glycerol) (DMPG) and 1,1′2,2′-tetra-oleoyl-cardiolipin (TOCL) were chosen. Powerful biophysical techniques were employed, including small-angle X-ray scattering (SAXS) and wide-angle X-ray scattering (WAXS), to understand the effect of BDQ on the nanostructure of the membrane models. The results showed that BDQ demonstrated a pronounced disordering effect in the bacterial cell membrane models, especially in the membrane model with cardiolipin (CL), while the human cell membrane model with large fractions of neutral phospholipids remained less affected. The membrane models and techniques provide detailed information about different aspects of the drug–membrane interaction, thus offering valuable information to better understand the effect of BDQ on their target membrane-associated enzyme as well as its side effects on the cardiovascular system.

## 1. Introduction

Tuberculosis (TB) is a major global health concern, with the disease being the leading cause of death worldwide among infectious diseases [1]. In 2018, the World Health Organization (WHO) reported an estimated 10.0 million new cases and 1.45 million deaths from TB. Significant progress has been made to reduce the global impact of TB, including drug discovery (WHO, 2019). In fact, there are several new drug candidates that are currently either in research or in clinical trials as well as several marketed drugs that are in a state of re-evaluation [2]. Bedaquiline (BDQ), a new drug that is also known as TMC207 (Figure 1), is a diarylquinoline compound that inhibits the bacterial ATP synthase located in the inner membrane [3]. BDQ was granted an accelerated approval by FDA in 2012 for the treatment of multi-drug-resistant TB (MDR-TB) in adults when other alternatives are not available [3,4]. However, BDQ has been associated with several adverse effects, including heart failure, which has raised concerns that its risks may outweigh its benefits [4]. Because BDQ must be deeply immersed in the membrane to reach its pharmacological target, we applied a biophysical approach to study the drug with the hypothesis that some of its adverse effects, especially the cardiovascular effects may indeed be membrane-mediated. To the best of our knowledge, this is the first time such a study has been carried out. In addition, it is well known that the efficacy of antibiotics depends on complex drug–membrane interactions. Indeed, antibiotic–cell membrane interactions play a crucial role in understanding the mechanism of action of drugs, their entry into the cellular compartments, and drug-induced toxicity [5]. In this context, the interaction of BDQ and membrane models was assessed using two biophysical techniques: small-angle X-ray scattering (SAXS) and wide-angle X-ray scattering (WAXS). The techniques were applied to obtain detailed information about the influence of this anti-TB drug on the structural organization of membrane models. These techniques are very powerful and allowed us to study drug interactions under biologically relevant conditions. These techniques gave important information about the changes in membrane organization induced by the drug. SAXS allowed us to study the effects of the drug on long-range bilayer organization, while WAXS was used to study changes in hydrocarbon chain packing during the interaction of the drug with lipid membranes.

To mimic prokaryotic and eukaryotic cell membranes, multilamellar vesicles (MLVs) of different phospholipid composition were chosen [5,6,7]. As phosphatidylcholines are one of the major components of plasmatic cell membranes, 1,2-dimyristoyl-*rac*-glycero-3-phosphocholine (DMPC), a zwitterionic phospholipid, was chosen as a simple eukaryotic cell membrane, mimicking the neutral charge of the surface membrane of eukaryotic plasma membranes [5]. In addition to phospholipids, cholesterol (CHOL) is also a major compound of the eukaryotic cell membrane and is fundamental for maintaining cell membrane structure as well as physiology and function of the membranes [8]. Thus, a 20% molar fraction of CHOL was chosen, with DMPC:CHOL (8:2 molar ratio) considered as a more reliable and biological relevant eukaryotic cell membrane model [9]. To mimic bacterial membrane, 1,2-dimyristoyl-*sn*-glycero-3-phospho-*rac*-(1-glycerol) (DMPG) was chosen because bacterial cell membranes predominantly contain negatively charged phospholipids. Cardiolipin (CL) is a key component to mimic bacterial cell membranes, and 1,1′2,2′-tetra-oleoyl-cardiolipin (TOCL) was therefore also selected as a model of the bacterial inner cell membrane [10,11]. 

The overall results of the study showed that BDQ had a pronounced disordering effect in the bacterial cell membrane models, especially in the membrane model with CL, while the human cell membrane model with large fractions of neutral phospholipids remained less affected. 

In summary, the main aim of this work was to contribute to a better understanding of drug–membrane interactions with the use of liposomes as membrane model systems and sophisticated biophysical techniques (SAXS and WAXS). The gathered results will lead to a better understanding of anti-TB drugs with a pharmacological target in the membrane, as is the case with the mechanism of action of BDQ, and will shed some light on their cardiovascular toxicity.

## 2. Materials and Methods

### 2.1. Materials

The lipids DMPC, CHOL, DMPG, and TOCL were purchased from Avanti Polar Lipids^®^, (Alabaster, AL, USA). The drug BDQ was purchased from AdooQ™ BioScience (Irvine, CA, USA). The buffer system (HEPES 0.01 M, pH = 7.4, I = 0.1 M) was prepared with ultrapure water produced by Millipore Milli-Q (conductivity 5.5 × 10^−6^ S/m), with the ionic strength adjusted with NaCl 0.1 M. Chloroform and methanol were purchased from Sigma-Aldrich (St. Louis, MO, USA).

### 2.2. Methods

#### 2.2.1. Multilamellar Vesicles

BDQ with drug:lipid was mixed with 9 mg of the lipids DMPC, DMPC:CHOL (8:2 molar ratio), DMPG, and TOCL (with 16 mM of Mg^2+^) in chloroform:methanol mixture (3:1 v/v) according to the required molar fraction of the drug. The lipid films were produced from these solutions and dried at 50.0 ± 0.1 °C under a stream of N_2_ and left overnight under reduced pressure to remove the organic solvents. Lipid films were hydrated, and the buffer system was added. The films were then alternately heated at 70.0 ± 0.1 °C, mixed by vortexing for about 5 min, and centrifuged for 30 s at 2000 g. The processes of vortexing and centrifugation were repeated three times. Lastly, the samples were aged overnight at 4.0 ± 0.1 °C and shaken by vortex at room temperature (21 °C) for 5 min. Subsequently, the dispersions were transferred into X-ray transparent glass capillaries of 1.5 mm diameter (Hilgenberg, Malsfeld, Germany). The flame-sealed capillaries were stored at 4.0 ± 0.1 °C until the SAXS and WAXS measurements.

#### 2.2.2. Small-Angle X-ray Scattering and Wide-Angle X-ray Scattering

SAXS and WAXS experiments were executed simultaneously at the Austrian SAXS beamline in the electron storage ring Elettra (Trieste, Italy) using a monochromatic radiation wavelength of 0.154 nm [12]. SAXS detector Pilatus3 1M (Dectris, Baden, Switzerland) was calibrated using silver behenate powder (*d*-spacing = 58.376 Å) and the WAXS detector Pilatus 100K (Dectris, Baden Switzerland) was calibrated with p-bromo benzoic acid, with the *d*-spacings taken from Ohkura et al. [13]. The static measurements were determined below and above the main phase transition temperature of the lipids using a custom-made sample cell placed in a water bath Unistat (Huber, Offenburg, Germany) for thermostatization. Each diffraction pattern was determined as described in previous studies by plotting the normalized scattering intensity in arbitrary units versus the magnitude of the scattering vector *s* in nm^−1^ [14].
$$s=\frac{4\pi sin\theta }{\lambda }$$
where θ is half of the scattering angle, and λ is the X-ray wavelength. The measured diffraction peaks were fitted with Lorentzians, and the positions of maximum intensities and full widths at half maximum were used to calculate the bilayer distances and the correlation lengths of the lipid bilayers, respectively. From the peak maximum positions (*s*) of the wide- and small-angle diffraction patterns, the lipid bilayer distances (*d*) were calculated as follows: $$d=\frac{2\pi }{s}$$

The correlation length (ξ) of the lipid bilayers was calculated from the full width at half maximum (*w*):
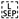

$$\xi =\frac{2\pi }{w}$$

## 3. Results and Discussion

### 3.1. Effects of BDQ on the Structure of DMPC Bilayers

Phosphatidylcholines, including DMPC, spontaneously form lamellar phases at ambient pressure and in excess of water, with the structure and long-range organization dependent on the temperature [14]. Static measurements at the temperatures of 10 and 37 °C were carried out by simultaneous SAXS and WAXS measurements to yield information on long-range bilayer organization and hydrocarbon chain packing, respectively [13]. The investigated temperatures of 10 and 37 °C were selected to cover the thermal ranges of the gel (Lβ’) and lamellar fluid (Lα) phases, respectively [13]. In Lβ’, the acyl chains are fully extended and packed in a distorted hexagonal lattice. The increment of the temperature leads to the main transition temperature (T_m_), where the acyl chains are conformationally disordered (*trans-gauche* isomerization) [13]. The repeat distance *d* deduced from the SAXS patterns as well as the distances between the polar head groups of DMPC obtained by WAXS measurements, including the correlation between the bilayers (ξ) of DMPC in the absence and in the presence of BDQ, are listed in Table 1. In the case of DMPC, the first-order Bragg reflection peaks on the scattering curves appeared at 2π/64.5, 2π/57.3 (10 °C), and 2π/62.3 (37 °C) Å^−1^; these values are in agreement with the literature [15]. The two first-order Bragg peaks with *d* values obtained for DMPC at 10 °C was indicative of the ripple phase (Pβ’) and not the expected gel phase (Lβ’). This was previously reported and may be explained by the short time of incubation of the DMPC samples, which did not allow the transition to the gel phase (Lβ’) that is characteristic at the temperature of 10 °C [15].

From Table 2, it is also possible to observe that the *d*-spacing for DMPC obtained at the temperature of 10 °C displayed a sharp reflection at 2π/4.21 Å^−1^, indicating the orthorhombic, hexagonal lattice of the chain packing arrangement that is characteristic of the ripple phase (Pβ’) [16]. At 37 °C, a diffuse reflection appeared in the Lα, corresponding to the disordered melted chain state (data not shown). The incorporation of BDQ in DMPC bilayers led to an almost negligible change in both SAXS and WAXS diffraction patterns. In the presence of BDQ, the first-order Bragg reflection peaks on the scattering curves appeared at 2π/65.8, 2π/55.6 (10 °C), and 2π/64.1 (37 °C) Å^−1^ (Table 1). Thus, BDQ slightly increased the bilayer thickness and the adjacent water layer of the DMPC, especially in the most biologically relevant mesophase (i.e., Lα phase). The increment of the *d*-spacings might have been due to an increment of the hydration of the phospholipid head groups. Furthermore, the correlation between the lipid bilayers (ξ) rose in the gel and in the fluid phases, pointing to an increase in the phospholipid molecular order. In the WAXS profile (Table 2), the *d*-spacings and the ξ were affected in a similar way to that observed in SAXS [17]. In fact, BDQ presented high lipophilicity with a predicted log*P* of 7.13 using MarvinView 16.10.10-0 from ChemAxon, meaning it probably penetrated deeply within the DMPC zwitterionic lipid bilayers.

### 3.2. Effects of BDQ on the Structure of DMPC:CHOL Bilayers

The obtained results indicate that, at the temperatures studied, the DMPC:CHOL 8:2 molar ratio mixture self-assembled into a lamellar phase. For the mixture of DMPC:CHOL 8:2 molar ratio, the first-order Bragg reflection peaks on the scattering curves appeared at 2π/69.7 (10 °C) and 2π/64.9 (37 °C) Å^−1^. In comparison with the d-values for pure DMPC bilayers (Table 1), the incorporation of 20 mol % of CHOL, as expected, led to a pronounced increase in bilayer thickness at all the temperatures studied [18]. This phenomenon can be explained by the decrease in the tilt angle of the phosphatidylcholines acyl chains and/or in the thickness of the hydration layer between the DMPC:CHOL bilayers. In fact, CHOL seemed to induce reorientation of the hydrocarbon chains of phospholipids, which became fully extended with a vertical orientation within the bilayer, resulting in a more ordered bilayer. Moreover, the presence of CHOL within phosphatidylcholine bilayers was responsible for an increment of the hydration layer due to the effect of CHOL as a spacer molecule, which allowed the water to further penetrate into the head group region [18]. The results in Table 3 reveal that the long-range distances of the DMPC:CHOL bilayer decreased with the temperature, whereas its correlation length decreased with increasing temperature. The decreasing of the cooperativity with the increment of the temperature was related to the disordering effect of temperature, which increased the spaces between the phospholipid and CHOL molecules, leading to a decrease in the molecular order [19]. BDQ led to a change in the SAXS and WAXS diffraction pattern of DMPC:CHOL in a more pronounced way in comparison with DMPC (Table 3 and Table 4). In the presence of BDQ, the first-order Bragg reflection peaks on the scattering curves appeared at 2π/69.5 (10 °C) and 2π/68.3 and 2π/66.1 (37 °C) Å^−1^ (Table 3). In the gel phase, despite the *d*-spacings remaining unalterable, the correlation length increased. In the fluid phase, the addition of BDQ led to an increment of the d-values obtained and an increase in the correlation length. The WAXS results (Table 4) showed that no significant alteration of the lipid bilayer occurred with the chosen concentration of BDQ. This effect might be due to the higher lipophilicity of BDQ and deep penetration into the phospholipid tail region without compromising the close packing of DMPC with CHOL molecules.

### 3.3. Effects of BDQ on the Structure of DMPG Bilayers

Negatively charged phospholipid DMPG at high salt concentrations exhibit a similar thermotropic behavior as that observed for phosphatidylcholines [20,21]. The SAXS diffraction patterns of DMPG showed a very diffuse and broad scattering, pointing to a lower correlation between the bilayers (data not shown), which is in full agreement with the literature. The repulsion between charged bilayers promoted the formation of large unilamellar vesicles (LUVs) instead of MLVs [22]. 

Table 5 displays the long-range repeat distances (*d*) and the correlation length (ξ) between the DMPG bilayers in the absence and in the presence of BDQ. The distances between the polar head groups of DMPG obtained by WAXS measurements in the absence and in the presence of BDQ are depicted in Table 6. From the analyses in Table 5, it can be observed that the Bragg reflection peaks on the scattering curves appeared at 2π/47.2 (10 °C) and 2π/47.8 (37 °C) Å^−1^. Due to the broad peaks obtained, the determination of the *d*-spacings was associated to a larger error bar (see Table 5). The *d*-spacings obtained are in accordance with the values described in the literature [22].

From the SAXS profile obtained, it was possible to observe that the incorporation of BDQ in the DMPG bilayers led to maintenance of the distances of the bilayer thickness. In addition, the water layer induced a disordering effect on the structure of the DMPG bilayers due to the pronounced changes in the correlation length (ξ) between the negatively charged bilayers. In the presence of BDQ, the Bragg reflection peaks on the scattering curves were centered at 2π/47.6 (10 °C) and 2π/47.8 (37 °C) Å^−1^ (Table 4). Thus, a decrease in the correlation length (ξ) in the gel phase and an increase in the fluid phase occurred for both concentrations. The significant alteration of the WAXS diffraction patterns proved the marked interaction of BDQ with the polar head groups of the negatively charged phospholipids. Although the distances between the phospholipid head groups remained unalterable, the overall decrease in the correlation length (ξ) supported the disturbance effect of this drug. Using MarvinView 16.10.10-0 from ChemAxon, it was predicted that BDQ at the physiological pH of 7.4 was a charged molecule with a significant contribution of positively charged molecules due to the protonation of the terminal amine group. Therefore, electrostatic interactions between the protonated amine and the deprotonated phosphate of DMPG might have played a key role in the pronounced interactions that occurred between the drug and the polar head groups [23]. 

### 3.4. Effects of BDQ on the Structure of TOCL Bilayers

Lipid TOCL at low salt concentrations forms bilayers with a lamellar phase pattern, with isotropic phases formed at high concentrations of salt [24,25]. In our experimental conditions (low concentration of salt), the SAXS diffraction patterns of TOCL showed a very diffuse and broad scattering, pointing to a lower correlation between the bilayers, which is full agreement with the literature (Table 7 and Figure 2). The distance between the polar head groups of TOCL obtained by WAXS measurements is depicted in Figure 3 and Table 8. It can be seen that the profile was very broad with no characteristic peak.

In fact, TOCL is composed of two negatively charged phosphate moieties, and repulsion between charged bilayers therefore probably promotes the formation of LUVs instead of MLVs. Interestingly, as can be observed from Figure 2 and Table 7, a completely different pattern of TOCL phase organization was formed in the presence of BDQ at 10 °C, with the occurrence of isotropic phases. In particular, several Bragg reflection peaks were located at 2π/98.2, 2π/71.4, 2π/63.5, and 2π/51.8 Å^−1^. Thus, the ratio obtained between the *d*-spacings (s1: $\sqrt{2}.$ s2: $\sqrt{3}.$ s3: 2. s4) was assumed to be a nonlamellar phase, attributed to a cubic structure (most probably Pn3m or Im3m). The establishment of electrostatic interactions as well as the hydrogen bonds between BDQ and the two phosphates of the TOCL may explain the presence of these structures and the several diffraction order peaks obtained (Figure 2 and Table 7). The repulsions between the negatively charged phosphate groups were minimized by the intercalation of the positively charged BDQ molecules, which acted as a shield of the surface net charge and restricted phosphate mobility. This reduced the head group area and promoted the tendency of forming nonlamellar phases. In agreement with the former observation, BDQ raised the correlation between the bilayers (ξ) and consequently the phospholipid order (Table 7).

At 37 °C, BDQ induced a pronounced decrease in the *d*-spacings, which can be explained by the smaller water layer between the bilayers (Table 7). Moreover, the WAXS profile obtained for BDQ (Figure 3) was consistent with the phospholipid order increment, and a single Bragg peak located at 2π/4.33 Å^−1^ was obtained only in the presence of the drug for both temperatures, reinforcing the strong interaction between BDQ and TOCL and the minimization of the electrostatic repulsions between the CL head groups (Table 8).

## 4. Conclusions

In this work, we studied for the first time the interactions of BDQ with eukaryotic and prokaryotic cell membrane models. In this context, SAXS and WAXS were used as two powerful biophysical techniques to assess drug–membrane interaction [14]. The results pointed to a differential interaction that was dependent on the charge of the head groups of the phospholipids and lipid composition. Thus, the effects of this novel anti-TB drug were almost negligible in the zwitterionic DMPC and DMPC:CHOL membrane models. In the case of plasma membrane models, BDQ, due its lipophilicity, established marked van der Waals forces, penetrating within the cell membranes without disturbing effects in the properties of the lipid bilayers. Recently, Belosludtsev et al. (2019) demonstrated that BDQ induced the formation of microdomains and permeabilization in DMPC liposomes as well as aggregation, shape change, and lysis of rat erythrocytes. The different results seem to be related to the different membrane models (MLVs instead of large unilamellar vesicles) and also the experimental conditions (pH, buffer, ratio between lipid:drug) [26]. The effects of this BDQ in membrane models were more evident in the negatively charged DMPG, especially in the TOCL membrane model. The more pronounced interactions of BDQ with the negatively charged lipid bilayers can be attributed to its positive charge, with the terminal amine group being responsible for the ionic interactions with the polar head groups of the negatively charged phospholipids. Thus, the mechanism of action of BDQ can also be related to their ability to destabilize the lamellar phases in the nonlamellar phases. This has already been described in the literature as an antibacterial mechanism of action related to membrane fusion and bacterial cell death through pore production in the membrane [27]. The adverse effects of BDQ can be related to its interaction with membranes. In fact, the main adverse effect of BDQ is related to cardiotoxicity, being the heart an organ rich in cardiolipins. Thus, the notorious cardiac toxicity of BDQ may be explained by the pronounced influence of BDQ in the heart muscle, which is rich in mitochondria. Mitochondrion is the major repository for cardiolipin in normal cells. Interestingly, this drug also demonstrated anticancer activity, with cancer cells showing significant accumulation of CL [28].

In conclusion, our findings support the hypothesis that the therapeutic activity of bedaquiline seems to be related to drug–membrane interactions. This reinforces the wider, emerging approach of studying the biophysical properties of membranes to determine the bioavailability and toxicity of drugs, ultimately allowing the design of more effective and less toxic drugs.

## Figures and Tables

**Figure 1 membranes-09-00141-f001:**
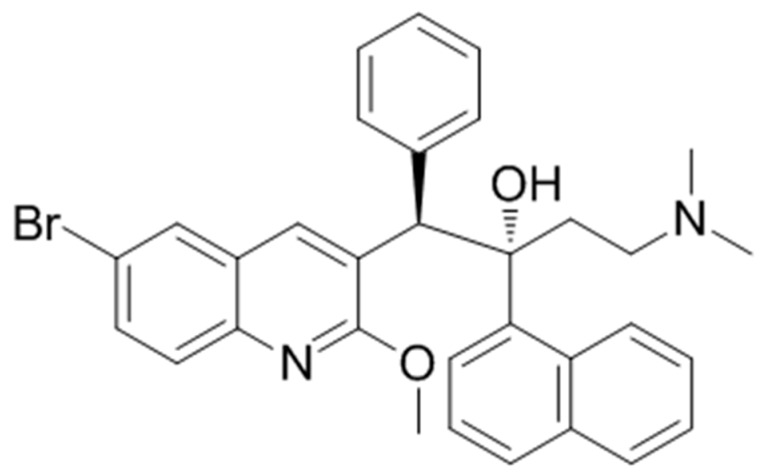
Chemical structure of bedaquiline (BDQ).

**Figure 2 membranes-09-00141-f002:**
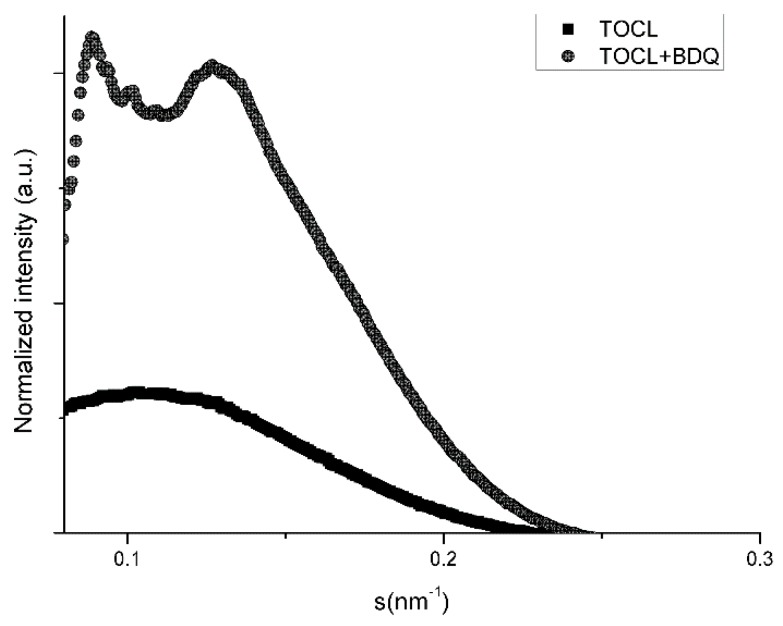
SAXS diffraction profiles of 1,1′2,2′-tetra-oleoyl-cardiolipin (TOCL) and TOCL with BDQ at 5% molar ratio at 10 °C.

**Figure 3 membranes-09-00141-f003:**
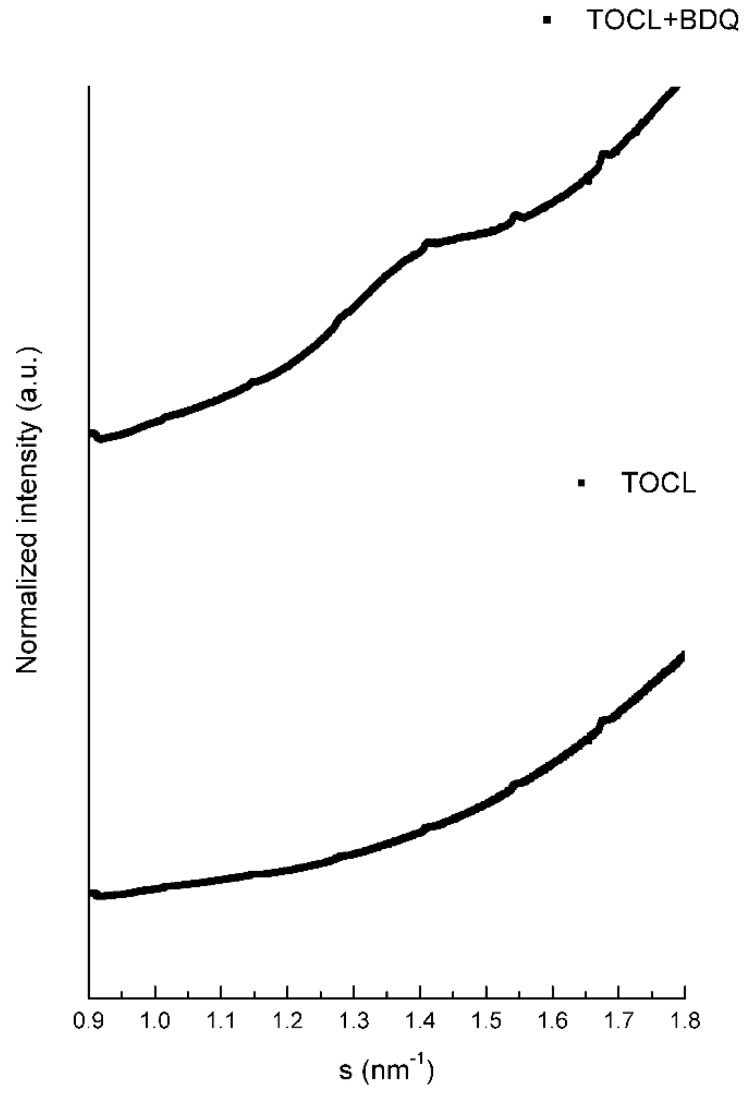
WAXS diffraction patterns of TOCL and TOCL with BDQ at 5% molar ratio at 10 °C.

**Table 1 membranes-09-00141-t001:** Small-angle X-ray scattering (SAXS) diffraction profiles of 1,2-dimyristoyl-*rac*-glycero-3-phosphocholine (DMPC) and DMPC with bedaquiline (BDQ) at 5% molar ratio at 10 and 37 °C.

Sample (mol %)	T (°C)	*d*_SAXS_ (Å)		ξ (Å)	
DMPC	10	64.5 ± 0.5	57.3 ± 0.5	697 ± 10	477 ± 10
	37	62.3 ± 0.5	-	629 ± 10	-
DMPC:BDQ	10	65.8 ± 0.5	55.6 ± 0.5	798 ± 10	855 ± 10
	37	64.1 ± 0.5	-	661 ± 10	-

**Table 2 membranes-09-00141-t002:** Wide-angle X-ray scattering (WAXS) diffraction patterns of DMPC and DMPC with BDQ at 5% molar ratio at 10 °C.

Sample	T (°C)	*d*_SAXS_ (Å)		ξ_1_ (Å)	ξ_2_ (Å)
		1st Peak	2nd Peak		
DMPC	10	4.2	4.0	767 ± 10	233 ± 10
DMPC:BDQ	10	4.2	4.1	967 ± 10	331 ± 10

**Table 3 membranes-09-00141-t003:** SAXS diffraction profiles of DMPC:cholesterol (CHOL) and DMPC:CHOL with BDQ at 5% molar ratio at 10 and 37 °C.

Sample	T (°C)	*d*_SAXS_ (Å)		ξ (Å)	
DMPC:CHOL	10	69.7 ± 0.5	-	966 ± 10	-
	37	64.9 ± 0.5	-	861 ± 10	-
DMPC:CHOL:BDQ	10	69.5 ± 0.5	-	1342 ± 10	-
	37	68.3 ± 0.5	66.1	731 ± 10	619

**Table 4 membranes-09-00141-t004:** WAXS diffraction patterns of DMPC:CHOL and DMPC:CHOL with BDQ at 5% molar ratio at 10 °C.

Sample	T (°C)	*d*_SAXS_ (Å)		ξ_1_ (Å)	ξ_2_ (Å)
		1st Peak	2nd Peak		
DMPC:CHOL	10	3.9	-	80 ± 10	-
DMPC:CHOL:BDQ	10	3.9	-	113 ± 10	-

**Table 5 membranes-09-00141-t005:** SAXS diffraction profiles of 1,2-dimyristoyl-*sn*-glycero-3-phospho-*rac*-(1-glycerol) (DMPG) and DMPG with BDQ at 5% molar ratio at 10 and 37 °C.

Sample	T (°C)	*d*_SAXS_ (Å)	ξ (Å)
DMPG	10	47.2 ± 0.5	314 ± 10
	37	47.8 ± 0.5	331 ± 10
DMPG:BDQ	10	47.6 ± 0.5	185 ± 10
	37	47.8 ± 0.5	370 ± 10

**Table 6 membranes-09-00141-t006:** WAXS diffraction patterns of DMPG and DMPG with BDQ at 5% molar ratio at 10 °C.

Sample	T (°C)	*d*_SAXS_ (Å)		ξ_1_ (Å)	ξ_2_ (Å)
		1st Peak	2nd Peak		
DMPG	10	4.03	3.86	132 ± 10	14 ± 10
DMPG:BDQ	10	4.06	3.89	60 ± 10	45 ± 10

**Table 7 membranes-09-00141-t007:** SAXS diffraction profiles of TOCL and TOCL with BDQ at 5% molar ratio at 10 and 37 °C.

Sample (mol %)	T (°C)	*d*_SAXS_ (Å)				ξ (Å)			
**TOCL**	10	60.3 ± 0.5	-	-	-	30 ± 10	-	-	-
	37	104.1 ± 0.5	-	-	-		35 ± 10	-	-
**TOCL:BDQ**	10	98.2 ± 0.5	71.4 ± 0.5	63.5 ± 0.5	51.8 ± 0.5	127 ± 10	776 ± 10	561 ± 10	137 ± 10
	37	76.9 ± 0.5	-	-	-	24 ± 10	-	-	-

**Table 8 membranes-09-00141-t008:** WAXS diffraction patterns of TOCL and TOCL with BDQ at 5% molar ratio at 10 and 37 °C.

Sample	T (°C)	*d*_SAXS_ (Å)	ξ_1_ (Å)	ξ_2_ (Å)
TOCL	10	NA	-	-
TOCL:BDQ	10	4.33	20 ± 10	-

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
