# Peer review of "Antituberculosis Drug Interactions with Membranes: A Biophysical Approach Applied to Bedaquiline"

_membranes, 2019, doi:10.3390/membranes9110141_

Round 1
Reviewer 1 Report
The study by Marina Pinheiro and colleagues, was carried out to a better understanding of drug- membrane interactions by the use of liposomes as membrane model systems by X-ray sophisticated biophysical techniques. In particular, they focused attention on the interaction of the anti-tuberculosis drug bedaquiline (BDQ) with different membrane models of eukaryotic and prokaryotic cell.
Since BDQ to reach its pharmacological target must be depth immersed in the membrane, the authors hypothesized that some of the adverse effects of this drug might be membrane-mediated. Indeed, their findings support the hypothesis that BDQ therapeutic activity seems to be related with the drug-membrane interactions, allowing in general the application of this kind of studies to design of more effective and less toxic drugs.
The experimental design was well thought out and all the assays were appropriately designed. Experiments are well conducted, and appropriate controls and statistical analyses are provided.
For all these reasons, the opinion of the referee is that the article, in its current form, is ready for publication.
Minor points:
-on line 22 there is an extra point;
- In table 1, in the first line, there are bold characters not united to the rest of the table;
- On line 238 there is an extra point;
- On line 238 there is an extra point;
- Table 7 is completely unformatted and there are uneven bold characters in the first line.
Author Response
Comment 1:
"The study by Marina Pinheiro and colleagues, was carried out to a better understanding of drug- membrane interactions by the use of liposomes as membrane model systems by X-ray sophisticated biophysical techniques. In particular, they focused attention on the interaction of the anti-tuberculosis drug bedaquiline (BDQ) with different membrane models of eukaryotic and prokaryotic cell.
Since BDQ to reach its pharmacological target must be depth immersed in the membrane, the authors hypothesized that some of the adverse effects of this drug might be membrane-mediated. Indeed, their findings support the hypothesis that BDQ therapeutic activity seems to be related with the drug-membrane interactions, allowing in general the application of this kind of studies to design of more effective and less toxic drugs.
The experimental design was well thought out and all the assays were appropriately designed. Experiments are well conducted, and appropriate controls and statistical analyses are provided.
For all these reasons, the opinion of the referee is that the article, in its current form, is ready for publication".
Thank you so much for this overall comment of the manuscript.
Comment 2:
"on line 22 there is an extra point".
Thank you so much. This is probably a formatting issue but in our last version no extra points exist.
Comment 3:
"In table 1, in the first line, there are bold characters not united to the rest of the table"
Thank you so much. This was corrected in the manuscript.
Comment 4:
"On line 238 there is an extra point"
Thank you so much.However, this is probably a formatting issue but in our last version no extra points exist.
Comment 5:
"Table 7 is completely unformatted and there are uneven bold characters in the first line".
Thank you so much. Once again this was corrected in the manuscript.
Reviewer 2 Report
Abstract:
The abstract has too many acronyms and does not facilitate reading and understanding. Line 19, CL is not defined.
Introduction:
The first sentence is not entirely true and is outdated. In addition, the reference provided is from 2013 and is not direct information from WHO. It is necessary to give more up-to-date, accurate and correct source information. (lines 27-28)
The subsequent paragraph refers to WHO data from 2015 (however, the reference is from 2013). (lines 31-33)
The work speaks of two powerful techniques for the study, but no description is given about them, their principles, the advantages of one over another and the reasons for its use in this study. That should be included in this introduction section.
one should use [()] for 1,2-dimyristoyl-sn-glycero-3-phospho-rac- (1-glycerol (DMPG). Line 61
Line 64, references should be unified as (10-11)
On Lines 65-67 an affirmation is made, and it is not clear whether it is a result of this study or if it is reported information, which is not referenced.
Materials and methods:
One of the most common water quality controls is electrical conductivity (S/m). I recommend changing to conductivity (line 78). Is the NaCl used from which source? It must be indicated (line 79) Chloroform and methanol used to prevent lipid bilayers must also be supported in the materials part. More detail on the development of liposomes should be given, indicating the volumes, times and agitation speeds used. This in order to provide accurate information that can be reproduced by other researchers. It is recommended to do this part in greater detail indicating step 1, step two, etc. I believe that a representative scheme of the assembly coupled to the principle and foundation of the technique would help a better understanding of how the experiment was carried out.
Results and Discussion:
The discussion of the results is focused only on the two techniques used. However, it is necessary to make a comparison with other types of techniques that are better well-known and that could also provide information about the membrane interaction process. Personally, it is hard for me to believe that only with these two techniques is there enough information to discuss such specific and detailed spatial arrangements and locations. I believe that for these statements it is necessary to compare with other techniques such as NMR in this solid or at least, to give more information on such techniques with extensive bibliographic reference.
Other aspects to improve:
Line 122, delete a point. Improve the resolution of figures 2 and 3, differentiating by colors. Organize the last columns of table 7.
Author Response
Comment 1:
"The abstract has too many acronyms and does not facilitate reading and understanding. Line 19, CL is not defined".
We agree with the referee. The abstract was checked and the acronyms were defined.
Comment 2:
"The first sentence is not entirely true and is outdated. In addition, the reference provided is from 2013 and is not direct information from WHO. It is necessary to give more up-to-date, accurate and correct source information. (lines 27-28)".
We are in agreement with the referee. The most recent epidemiological results from WHO were added to the manuscript, namely from the report of 2019.
Comment 3:
"The subsequent paragraph refers to WHO data from 2015 (however, the reference is from 2013). (lines 31-33)".
The reference was replaced and the epidemiological data was taken by the Gobal Tuberculosis Report from WHO, 2019 that reports the data from the previously year, 2018.
Comment 4:
"The work speaks of two powerful techniques for the study, but no description is given about them, their principles, the advantages of one over another and the reasons for its use in this study. That should be included in this introduction section".
We understand the referee concern. Both techniques are complementary, giving different information. According the recommendation we included new information in the introduction.
New information inserted:
Line 49
These techniques are very powerful, allowing to study the interactions of drugs under biologically relevant conditions. These techniques give commentary information about the changes on membrane organization induced by the drug. SAXS allow to study the effects of drug on long-range bilayer organization, while WAXS to study changes in hydrocarbon chain packing during the interaction of the drug with lipid membranes.
Comment 5:
"one should use [()] for 1,2-dimyristoyl-sn-glycero-3-phospho-rac- (1-glycerol (DMPG). Line 61".
We acknowledge the referee recommendation and we use [()] for 1,2-dimyristoyl-sn-glycero-3-phospho-rac- (1-glycerol (DMPG).
Comment 6:
"Line 64, references should be unified as (10-11)".
We acknowledge the referee correction and the references were unified.
Comment 7:
"On Lines 65-67 an affirmation is made, and it is not clear whether it is a result of this study or if it is reported information, which is not referenced".
This is the "overall results" of our study. To be more clear we replaced the sentence.
New information inserted:
Line 69
"The overall results present in this study demonstrated that BDQ shown a pronounced disordering effect in the bacterial cell membrane models, especially in the membrane model with CL, while the human cell membrane model with large fractions of neutral phospholipids remains less affected".
Comment 8:
"One of the most common water quality controls is electrical conductivity (S/m). I recommend changing to conductivity (line 78). Is the NaCl used from which source? It must be indicated (line 79) Chloroform and methanol used to prevent lipid bilayers must also be supported in the materials part. More detail on the development of liposomes should be given, indicating the volumes, times and agitation speeds used. This in order to provide accurate information that can be reproduced by other researchers. It is recommended to do this part in greater detail indicating step 1, step two, etc. I believe that a representative scheme of the assembly coupled to the principle and foundation of the technique would help a better understanding of how the experiment was carried out".
We agree with the referee and more information was given in the material/methods of the manuscript to allow replicability by other researchers.
Comment 9:
"The discussion of the results is focused only on the two techniques used. However, it is necessary to make a comparison with other types of techniques that are better well-known and that could also provide information about the membrane interaction process. Personally, it is hard for me to believe that only with these two techniques is there enough information to discuss such specific and detailed spatial arrangements and locations. I believe that for these statements it is necessary to compare with other techniques such as NMR in this solid or at least, to give more information on such techniques with extensive bibliographic reference".
We understand the referee concern. This drug, BDQ is a new drug, recently in the market and for that reason few studies exist. Indeed, only one biophysical study of the interaction between BDQ and membrane models exist and is recent (2019) and the results were inserted, compared and discussed in the manuscript.
Comment 10:
"Line 122, delete a point. Improve the resolution of figures 2 and 3, differentiating by colors. Organize the last columns of table 7".
We agree with the referee and the point was deleted and the Table 7 was replaced.
Reviewer 3 Report
The present manuscript describes the interaction of the anti-tuberculosis drug bedaquiline (BDQ) with membrane models composed of DMPC, CHOL, DMPG and cardiolipin. The lipid mixture is therefore complex and biologically relevant in the biophysical field. The samples were characterized by SAXS and WAXS. The gel and lamellar fluid phases were probed by including 10 ºC and 37 ºC. The impact of the drug on the structure of the membrane models has been described in great detail. The manuscript reads well and the experiments are sound.
I would recommend the publication of the manuscript after the following minor revision:
* “In comparison with the d-values for pure DMPC bilayers (Table 1), the incorporation of 20 mol% of CHOL leads to a pronounced increase in bilayer thickness at all the temperatures studied.” Could the authors elaborate on this point? Was this increase in thickness of a similar value to previously published reports?
* “The distances between the polar head groups of DMPG obtained by WAXS measurements in the absence and in the presence of the BDQ are depicted in Table 6.” What would be the value of area per DMPG lipid?
* “Thus, only the electrostatic interactions between the BDQ and the two phosphates of the TOCL explain the presence of these structures, regarding the several diffraction order peaks obtained (Figure 2) (Table 7)” Is it possible to consider the formation of H-bonds herein?
Author Response
Comment 1:
"The present manuscript describes the interaction of the anti-tuberculosis drug bedaquiline (BDQ) with membrane models composed of DMPC, CHOL, DMPG and cardiolipin. The lipid mixture is therefore complex and biologically relevant in the biophysical field. The samples were characterized by SAXS and WAXS. The gel and lamellar fluid phases were probed by including 10 ºC and 37 ºC. The impact of the drug on the structure of the membrane models has been described in great detail. The manuscript reads well and the experiments are sound".
Thank you so much for his overall comment.
Comment 2:
“In comparison with the d-values for pure DMPC bilayers (Table 1), the incorporation of 20 mol% of CHOL leads to a pronounced increase in bilayer thickness at all the temperatures studied.” Could the authors elaborate on this point? Was this increase in thickness of a similar value to previously published reports?".
We acknowledge the recommendation. In fact, this phenomena is already described in the literature and we add more information and a proper reference to the sentence.
Before corrections:
In comparison with the d-values for pure DMPC bilayers (Table 1), the incorporation of 20 mol% of CHOL leads to a pronounced increase in bilayer thickness at all the temperatures studied.
After corrections:
In comparison with the d-values for pure DMPC bilayers (Table 1), the incorporation of 20 mol% of CHOL as expected leads to a pronounced increase in bilayer thickness at all the temperatures studied (18).
Comment 3:
“The distances between the polar head groups of DMPG obtained by WAXS measurements in the absence and in the presence of the BDQ are depicted in Table 6.” What would be the value of area per DMPG lipid?".
The values for DMPG are also depicted in the Table 6. Thus, for DMPG the values are 4.03 and 3.86 angstrom.
Comment 4:
“Thus, only the electrostatic interactions between the BDQ and the two phosphates of the TOCL explain the presence of these structures, regarding the several diffraction order peaks obtained (Figure 2) (Table 7)” Is it possible to consider the formation of H-bonds herein?".
We are in full agreement with the referee. In fact, the formation of H-bonds can also explains this phenomena and for that reason the authors acknowledge and add this important information to the manuscript.
Round 2
Reviewer 2 Report
Each of the suggestions were taken into account